# Differences in Glycoproteins and the Potential for Early Protection Using LAIV Based on Drift Variants of the A/H1N1pdm09 Influenza Virus

**DOI:** 10.3390/vaccines13090966

**Published:** 2025-09-11

**Authors:** Yulia Desheva, Irina Mayorova, Andrey Rekstin, Daniil Sokolovsky, Polina Kudar, Nina Kopylova, Danila Guzenkov, Darya Petrachkova, Andrey Mamontov, Andrey Trullioff, Irina Kiseleva

**Affiliations:** 1Federal State Budgetary Scientific Institution “Institute of Experimental Medicine”, 197022 St. Petersburg, Russia; st076107@student.spbu.ru (I.M.); arekstin@yandex.ru (A.R.); dan1fy@yandex.ru (D.S.); polina6226@mail.ru (P.K.); ninzinya@mail.ru (N.K.); danila.guzenkov@yandex.ru (D.G.); ya.dashook@ya.ru (D.P.); an.mamontow@yandex.ru (A.M.); andrey@iem.sp.ru (A.T.); irina.v.kiseleva@mail.ru (I.K.); 2Medical Institute, St. Petersburg State University, 199034 St. Petersburg, Russia

**Keywords:** live attenuated influenza vaccine, antigenic drift, A(H1N1)pdm09, hemagglutinin, neuraminidase, glycosylation, early cytokines, type I interferon, MX1

## Abstract

Background/Objectives: Antigenic drift of influenza A(H1N1pdm09) viruses has led to periodic replacement of vaccine strains. Understanding how structural differences in glycoproteins influence immune protection is crucial for improving vaccine effectiveness. Methods: We conducted a structural analysis of the hemagglutinin (HA) and neuraminidase (NA) glycoproteins from drifted A(H1N1)pdm09 strains: A/South Africa/3626/2008 and A/Guangdong–Maonan/SWL1/2020, as well as their cold-adapted live attenuated vaccine (LAIV) reassortant strains (A/17/South Africa/2013/01(H1N1)pdm09 and A/17/Guangdong–Maonan/2019/211(H1N1)pdm09). We compared their replication in chicken embryo and mammalian cell culture, assessed type I interferon induction, and evaluated post-vaccine protection in mice after homologous and heterogeneous viral challenges. Results: The two vaccine strains had distinct glycosylation patterns for HA and NA. However, they had similar replication capacity in embryonated egg and mammalian cells. In the mouse respiratory tract, both strains replicated similarly. A/17/South Africa/2013/01(H1N1)pdm09 induced significantly higher levels of IFN-α and Mx1 in vitro, and it elicited earlier IgM and IgG response after vaccination in mice. At day 6 after immunization, it provided 70% protection from homologous challenge. A/17/Guangdong–Maonan/2019/211(H1N1)pdm09 did not prevent death, but it reduced viral titer in the lungs. Interestingly, A/17/South Africa/2013/01(H1N1)pdm09 provided full protection from heterologous H5N1 challenge, while A/17/Guangdong–Maonan/2019/211(H1N1)pdm09) only provided partial protection. Conclusions: Differences in HA and NA glycans among A(H1N1)pdm09 strains may influence innate and adaptive immunity, as well as cross-protection. These findings emphasize the importance of glycoprotein structure when selecting vaccine candidates for optimal homologous and cross-protection against influenza.

## 1. Introduction

Influenza A viruses (IAVs), which cause acute respiratory illness and are responsible for millions of deaths annually, remain a major global health concern. Vaccination is therefore a priority in medical and biological research. Antigenic drift can compromise the effectiveness of influenza vaccines; however, this limitation applies, to a lesser extent, to live attenuated influenza vaccines (LAIVs). When administered intranasally, LAIV initiates innate immune responses from the earliest days post-vaccination, including the production of type I interferons, early cytokines, and poly-specific IgM antibodies with strain-specific activity [1]. These early responses pre-stimulate the immune system and can significantly reduce the replication of respiratory viruses, making LAIV particularly relevant under current epidemiological conditions.

LAIV has been extensively studied and used in multiple countries over several decades, with a well-established safety and efficacy profile. Clinical trials have demonstrated that LAIV effectively prevents influenza infection and reduces disease severity in vulnerable populations [2]. LAIV is also capable of inducing robust T-cell responses [3]. Compared with inactivated vaccines, intranasal immunization with LAIV elicits a more natural immune response [4], characterized by both systemic antibody production and the generation of long-lived memory cells [5]. In addition, secretory IgA induced in the respiratory tract provides local mucosal protection at the primary entry site of infection [6]. Trivalent LAIVs derived from donor attenuated strains A/Leningrad/134/17/57 and B/USSR/60/69 were first licensed in the USSR in 1987 and remain in use for adults and children from the age of three [7]. In the United States, donor strains A/Ann Arbor/6/60 and B/Ann Arbor/1/66 were licensed in 2003, having been generated through serial passage of parental epidemic strains in chicken embryo kidney cells under progressively reduced temperatures [8]. Based on WHO recommendations, the American LAIV has been produced as a quadrivalent vaccine since 2012, covering both influenza B lineages (Yamagata and Victoria). In 2023, however, the WHO recommended returning to trivalent vaccines following the disappearance of B/Yamagata from circulation [9].

Beyond influenza-specific protection, live vaccines can also provide non-specific benefits, including broad cross-protection against unrelated reinfections and potential reduction in certain autoimmune processes [10]. Recent work attributes these effects to mechanisms such as innate immune memory, or “trained immunity,” as well as heterologous T-cell responses. Trained immunity refers to epigenetic and metabolic reprogramming of innate immune cells—including monocytes, macrophages, NK cells, and their progenitors—which enhances their responsiveness to subsequent homologous or heterologous stimuli [11]. Harnessing these non-specific vaccine effects may aid in the development of universal vaccines targeting a broad spectrum of pathogens.

IAV surface glycoproteins hemagglutinin (HA) and neuraminidase (NA) are N-glycosylated. In mammalian proteins, N-linked glycosylation typically occurs at Asn residues within the Asn-X-Ser/Thr (X ≠ Pro) motif [12]. However, motif presence alone does not guarantee glycosylation; occupancy depends on local sequence context, protein conformation, and accessibility during ER/Golgi processing, with exceptions to the canonical rule reported [13,14]. Glycan structures on HA/NA differ by host cell type and intracellular processing and may include high-mannose, hybrid, or complex glycans. HA glycans may undergo sulfation, while sialic acids on complex glycans are often cleaved by NA [15]. These variations influence viral virulence, immune evasion, and cytokine responses, thereby shaping host protection. NA further modulates host immunity by regulating CD8^+^ T-cell activation, and LAIV expressing WSN-NA has been shown to act as an effective vaccine [16].

In seasonal influenza vaccination, a full immune response typically develops within 2–4 weeks [17], although exposure may occur earlier. Vaccination during outbreaks of acute respiratory infections is also feasible. Importantly, LAIV—unlike inactivated vaccines—has been shown to interrupt outbreaks indirectly [18]. In murine models, cold-adapted LAIV elicited rapid innate responses in the respiratory tract, limiting replication of respiratory syncytial virus (RSV) and blunting early proinflammatory cytokine surges following RSV challenge [10].

Our earlier studies demonstrated that seasonal LAIV conferred protection against lethal influenza challenge, and even against secondary streptococcal infection, within the first days post-vaccination in murine models [1,19,20,21]. These effects correlated with increased early cytokine production in the respiratory tract and accelerated responses upon viral challenge. The role of innate immunity during LAIV immunization is particularly important for populations with diminished adaptive immunity, such as the elderly, in whom innate mechanisms may partially compensate for reduced T- and B-cell responses [22].

Early induction of type I interferons and related cytokines is a key determinant of immediate antiviral defense, restricting viral replication before adaptive responses mature [23]. Myxovirus resistance protein 1 (MX1), an interferon-inducible GTPase that disrupts influenza virus replication by targeting nucleoprotein and inhibiting transcription, serves as both an effector and a biomarker of interferon-mediated antiviral states [24]. LAIV constructs engineered to enhance interferon responses, such as NS1-deficient strains, demonstrate accelerated innate activation and pre-adaptive protection [25]. Notably, conventional cold-adapted LAIVs with intact NS1 can still elicit strong local innate responses, suggesting that surface glycoprotein architecture and other viral features also shape early immune activation [26].

In February 2020, the World Health Organization recommended inclusion of the A/Guangdong–Maonan/SWL1536/2019(H1N1)pdm09-like virus in the 2020–2021 Northern Hemisphere vaccine composition. Many contemporary A(H1N1)pdm09 isolates of subclade 6B.1A5A carried substitutions in known antigenic sites and the 190-helix, altering both antigenicity and N-glycosylation motifs on HA [27,28]. These observations motivated direct comparison of A/Guangdong–Maonan with the earlier A/South Africa/3626/2013 strain, as differences in HA/NA glycosylation and structure could modulate innate recognition and early LAIV-mediated protection.

In this study, we investigated whether structural differences in HA and NA between two A(H1N1)pdm09 LAIV candidates—A/South Africa/3626/2013 and A/Guangdong–Maonan/SWL1536/2019—influence early innate activation, humoral kinetics, and protective efficacy in a murine model. To this end, we integrated in silico analyses of HA/NA glycosylation with in vitro replication and cytokine assays, as well as in vivo experiments evaluating replication, antibody kinetics, and protection against homologous and heterologous challenge following a single intranasal immunization.

## 2. Materials and Methods

### 2.1. Viruses

The following influenza A virus strains were used in the study: the A/17/South Africa/2013/01(H1N1)pdm09 and A/17/Guangdong–Maonan/2019/211 (H1N1)pdm09. The candidate vaccine strains used in this study were reassortants that inherited hemagglutinin (HA) and neuraminidase (NA) from A/South Africa/3626/13 (H1N1)pdm09 or A/Guangdong–Maonan/SWL1536/2019(H1N1)pdm09 influenza virus, respectively, and PB2, PB1, PA, M, and NS genes from A/Leningrad/134/17/57(H2N2) master donor strain (MDS). For experimental infection of laboratory animals, the pandemic influenza virus A/California/09/07(H1N1)pdm09-MA (MA—mouse-adapted) was used, kindly provided by the head of the laboratory of chemotherapy of viral infections of the A.A. Smorodintcev Research Institute of Influenza of the Ministry of Health of the Russian Federation, Research Institute of Cand. of Biological Sciences A.A. Shtro. infecting virus A/Indonesia/5/2005(H5N1) IDCDC-RG2 was produced by reverse genetics by the National Institute of Biological Standards and Control (NIBSC, Hertfordshire, United Kingdom) [29].

Viruses were grown in the allantoic cavity of 10-day-old embryonated hen’s eggs at the temperature of 32 °C for 48 h. Infectious allantoic fluid was divided into aliquots and stored at −70 °C. Fifty percent egg infectious dose (EID50) titers were determined by titration of viruses in eggs and calculated by Reed and Muench method [30].

### 2.2. Molecular Genetic Analysis

The translation of HA sequences from influenza viruses A/17/South Africa/2013/01(H1N1)pdm09 and A/17/Guangdong–Maonan/2019/211 (H1N1)pdm09 was performed using an online tool ExPASy Translate Tool (https://web.expasy.org/translate/, accessed on 30 April 2025) [31].

Automatic prediction of glycosylation sites was performed using NetNGlyc 1.0 Server service [32]. Sites with a confidence level ≥ 7 out of 9 (“jury agreement”) were taken into account.

Models of NA and HA protein were created by homologous modeling using the online platform AlphaFold Server (https://alphafoldserver.com/, accessed on 30 April 2025) [33].

Visualization of the 3D structures and analysis of the locations of glycosylation site locations were performed in PyMOL (version 3.0.3, Schrödinger, LLC, New York, USA). Predictiona of N-glycans were shown as red highlights on the surface HA molecule and yellow, red, and green highlights on NA.

### 2.3. Cell Lines

MDCK cells were seeded on a 96-well plate at a density of 3.5 × 10^6^ per ml. On the next day, the maintenance medium was removed, the cells were washed twice with sterile PBS, and then 100 μL of the virus-containing medium without FBS but with the addition of Trypsin-TPCK (5 μg/mL, Sigma, St. Louis, MO, USA) were added to each well. Cells were incubated at 33 °C for 72 h.

A-549 cells lung carcinoma cells were obtained from the Cell Culture Bank of the Cytology Institute, Russian Academy of Sciences, St. Petersburg, Russia, and were cultured in Dulbecco’s Modified Eagle Medium (DMEM, Biolot, St. Petersburg, Russia) supplemented with 10% of inactivated fetal calf serum (Capricorn Scientific, Ebsdorfergrund, Germany), gentamicin 50 μg/mL (Biolot, St. Petersburg, Russia), and 2 mM L-Glutamine (BioLot, St. Petersburg, Russia). Cells were cultured in plastic flasks of 50 mL (Sarstedt, Nümbrecht, Germany). They were incubated at 37 °C in a 5% CO_2_ atmosphere. For experiments, the cells were plated in 24-well flat-bottom plates (Sarstedt, Nümbrecht, Germany) and incubated until a confluent monolayer was formed.

THP-1, human alveolar macrophages(ATCC TIB-202) were used to study cytokinae production. THP1 cells were cultured in RPMI 1640 medium supplemented with fetal calf serum (10%), antibiotics (2 mM L-glutamate), non-essential amino acid (0.1 mM), and sodium pyruvates (1.0 mM).

To study viral cytokine profiles, approximately 1 × 10^6^ per mL of A549 or THP-1 cells were infected with virus at a multiplicity of infection (MOI) of 0.1 for 1 h, ten washed and cultured for 24 h in 24-well plates. As a positive control, we used the Toll-like receptor agonist polyinosinic:polycytidylic acid (Poly I:C) (Sigma, St. Louis, MO, USA) at a final concentration of 1 μg/mL. Medium alone was used as a control. Supernatants were harvested and stored at the –70 °C. Levels of TNF-α, IL-6, MCP-1 and IFN-α in supernatants were measured using commercial ELISA kits (Vector-Best, Novosibirsk, Russia) according to the manufacturer’s instructions. Myxovirus Resistance 1 (MX1) protein levels were measured by ELISA using a SEL763Hu ELISA Kit (Cloud-Clone, Wuhan, China).

### 2.4. Mice

Female CBA mice aged 6–8 weeks were supplied by the laboratory breeding nursery “Rappolovo” (St. Petersburg Region, Russia) and were selected for this study because this strain has been used consistently in previous LAIV and cold-adapted influenza studies and displays reproducible susceptibility and immune responses to mouse-adapted A(H1N1)pdm09 and related challenge strains [34]. Female mice were chosen to reduce variability due to inter-male aggression and associated cage stress, which can confound weight loss and immune readouts in group-housed male cohorts [35]. The 6–8-week age window corresponds to young adult mice with a mature, yet immunologically naive, phenotype commonly used in vaccine efficacy and immunogenicity studies; this age range minimizes developmental variability while remaining broadly comparable to prior vaccine studies [36].

### 2.5. Ethics Approval

All proce5dures for the use and care of animals were approved by the Local Bioethical Committee of the Institute of Experimental Medicine (No. 3/23 from 20 September 2023).

### 2.6. Viral Replication in Mice

To assess viral infectivity, mice were lightly anesthetized with ether and inoculated intranasally with 50 µL of phosphate-buffered saline (PBS) containing 6.0 log10 EID50 of each test virus, administered equally into both nostrils. Viral loads were quantified in upper and lower respiratory tract tissues collected on day 3 post-immunization. Tissue samples were homogenized in 1.0 mL of sterile PBS supplemented with antibiotic–antimycotic (Invitrogen, Paisley, UK) using a TissueLyser LT bead mill (QIAGEN, Hilden, Germany). Clarified supernatants were subsequently titrated in embryonated chicken eggs under conditions appropriate for determining viral titers.

### 2.7. Immunization and Early Protection Activity Study

Mice were randomly assigned to three groups (20 animals per group). Two groups were intranasally immunized under light ether anesthesia with virus-containing allantoic fluid (diluted in PBS) of either LAIV SA or LAIV Maonan at a dose of 6.0 log10 EID50. The third group served as a control and received 50 µL of PBS intranasally. The median lethal dose (MLD50) values of the challenge viruses were determined in preliminary titration experiments in naïve CBA mice. Groups of five animals were intranasally inoculated with ten-fold serial dilutions of virus (50 µL per mouse) and monitored daily for 14 days. The MLD50 was calculated using the Reed–Muench method [30]. On day 6 post-immunization, mice from each group were intranasally challenged under light ether anesthesia with either pandemic influenza virus A/California/09/07 (H1N1)pdm09-MA at a dose of 100 MLD50 or avian influenza virus A/Indonesia/5/2005 (H5N1) IDCDC-RG2 at a dose of 1 MLD50. On day 3 post-challenge, lungs from five mice in each group were collected to assess replication of the wild-type viruses. The remaining 10 mice per group were monitored for 10 days to evaluate body-weight loss and mortality.

### 2.8. Sample Collection and Antibody Detection

Blood samples were collected from the retro-orbital sinus of mice (*n* = 6 per group) under light ether anesthesia on day 6 after immunization and subsequently at weekly intervals up to 3 weeks. Sera were separated by centrifugation at 1500× *g* for 10 min and stored at −20 °C until use. For the determination of local antibodies, saliva/nasal wash samples were obtained from 6 mice per group after intraperitoneal injection of 0.1 mL of 0.5% pilocarpine solution. Samples were collected into tubes containing 0.001 M phenylmethylsulfonyl fluoride (PMSF) to inhibit protease activity and stored at −20 °C until analysis. Serum IgM and IgG, as well as mucosal IgA antibody responses, were measured by enzyme-linked immunosorbent assay (ELISA). Briefly, 96-well ELISA plates (Sarstedt, Nümbrecht, Germany) were coated overnight at 4 °C with whole purified A/California/09/07(H1N1)pdm09 virus (20 hemagglutination units, HAU per well). After blocking with 1% BSA in PBS, serial two-fold dilutions of serum or mucosal samples were added. Bound antibodies were detected with horseradish peroxidase-conjugated anti-mouse IgM, IgG, or IgA secondary antibodies (Sigma-Aldrich, St. Louis, MO, USA) and TMB substrate (Sigma-Aldrich, St. Louis, MO, USA). Optical density (OD) was measured at 450 nm. Final antibody titers were expressed as the highest dilution at which OD450 exceeded the mean OD450 of negative-control wells plus three standard deviations. Antibody titers were expressed as geometric mean titers (GMTs) and log2-transformed for statistical analysis.

### 2.9. Statistical Analysis

The observed values of the experiments were expressed as the means ± standard deviation (SDs). Antibody titers were expressed as geometric mean titer (GMT) and for statistical analysis were expressed as log2. Differences between two groups were determined using a nonparametric test, the Wilcoxon–Mann–Whitney test, and when comparing three or more groups statistical significance was determined using Kruskall–Wallis tests with Dunn’s multiple comparison in Prism software v9.0 (GraphPad software Inc., San Diego, CA, USA), with *p* < 0.05 being considered statistically significant.

## 3. Results

### 3.1. Theoretical Analysis of Glycosylation in Two Surface Proteins of the Influenza A/H1N1pdm09 Virus

We analyzed potential glycosylation sites in HA and NA of A/17/South Africa/2013/01(H1N1)pdm09 and A/17/Guangdong–Maonan/2019/211(H1N1)pdm09 viruses (Figure 1 and Figure 2, Table 1 and Table 2).

It was shown that a N179 substitution associated with the appearance of a new glycosylation site has been found in the HA of the A/17/Guangdong–Maonan/2019/211(H1N1)pdm09 strain, which was not present in the A/17/South Africa/2013/01(H1N1)pdm09 HA.

The disappearance of the glycosylation site at position 50 in the NA stem (NQSV), and a possible increased probability of glycosylation at Asn^42^, are shown in the vaccine strain A/17/Guangdong–Maonan/2019(H1N1)pdm09.

### 3.2. Reproduction of Vaccine Viruses in Chicken Eggs and MDCK Cell Culture

As can be seen in Figure 3, both vaccine viruses, A/17/South Africa/2013/01(H1N1) pdm09 and A/17/Guangdong–Maonan/2019/211(H1N1) pdm09, reproduced equally well at the optimal temperature in chicken embryos and MDCK cells, despite differences in their surface proteins.

We conducted a study on the cytokine profile of viruses, examining the production of type I interferon and MX1 protein in A549 cell culture. It was shown that the A/17/South Africa/2013/01(H1N1)pdm09 vaccine virus induces a marked production of interferon I and the MX1 protein, in epithelial cells (Figure 4a,b).

In addition, the A/17/South Africa/2013/01(H1N1)pdm09 vaccine strain and its parent ‘wild-type’ virus A/South Africa/3626/13 (H1N1)pdm09 induce the most significant production of TNF-α and on 24 h after incubation with A549 cultures (Figure 4c).With respect to IL-6, both the A/17/South Africa/2013/01(H1N1)pdm09 and A/17/Guangdong–Maonan/2019/211(H1N1)pdm09 vaccine strains induce statistically significantly higher levels of production of this cytokine than control unstimulated cells (Figure 4d). As for MCP-1 levels, only the A/South Africa/3626/13 (H1N1)pdm09 virus caused a significant increase in the production of this chemokine compared to the unstimulated control (Figure 4e).

In cells of monocyte–macrophage origin, it has been shown that these cell type, when exposed to the influenza virus, produce significantly interferon type I than A549 cells. However, the MX protein is not detected (Figure 5a,b).

In THP-1 cells, influenza viruses induced higher levels of TNF-α and lower levels of IL-6 compared with A549 cells (Figure 5c,d). The MCP-1 levels were significantly elevated after exposure to A/South Africa/3626/13(H1N1)pdm09 and A/17/South Africa/2013/01(H1N1)pdm09 (Figure 5e).

### 3.3. Reproduction of Vaccine Viruses in the Respiratory Tract of Mice and Immunogenicity

Reproduction of the vaccine strain A/17/South Africa/3626/13(H1N1)pdm09 in the lungs of mice, as determined by titration in chicken embryos, was higher than that of A/17/Guangdong–Maonan/2019/211(H1N1)pdm09 on the 3rd day after immunization (*p* = 0.0019, Kruskal–Wallis test), although, pairwise comparisons showed no differences between individual groups (Figure 6a). The reproduction of neither vaccine strain was detected in the nasal passages (Figure 6a).

Determination of the viral load in the organs of immunized mice by the rRT-PCR method did not reveal statistically significant differences (*p* = 0.013, Kruskal–Wallis test; Figure 6b). A study of cytokine production in the organs on day 3 post-immunization also did not show any significant differences between vaccine strains (Figure 6b,c). However, the vaccine strain A/17/South Africa/3626/13(H1N1)pdm09 caused a more pronounced induction of cytokines, especially type I interferons, except for IL-6, in the lungs compared to other strains. The formation of antibodies after vaccination with two strains was studied. On the sixth day after vaccination, the levels of specific IgM and IgGs were significantly higher after A/17/South Africa/3626/13(H1N1)pdm09 immunization than after A/17/Guangdong–Maonan/2019/211(H1N1)pdm09 immunization (Figure 7a). Local IgA levels were also slightly higher in vaccinated mice, but this difference was not statistically significant (*p* = 0.5, Kruskal–Wallis test; Figure 7a).

A study of the dynamics of serum antibody formation showed that immunization with LAIV A/17/South Africa/2013/01(H1N1)pdm09 caused the formation of IgG in serum to be statistically significantly higher compared to immunization with a vaccine strain of A/17/Guangdong–Maonan/2019/211(H1N1)pdm09 (Figure 7b).

### 3.4. Protection Against Homologous and Heterologous Influenza Infection

An experiment on mice showed that intranasal immunization with a LAIV based on the A/17/Guangdong–Maonan/2019/211(H1N1)pdm09 strain did not provide protection against lethal infection with the mouse-adapted A/H1N1pdm09 virus on day 6 day after immunization (Figure 8a). At the same time, a LAIV based on A/17/South Africa/2013/01(H1N1)pdm09 protected 70 percent of animals from death (Figure 8a). The weight curves differed slightly (Figure 8b), but the viral load was statistically significantly reduced after immunization with LAIV based on A/17/Guangdong–Maonan/2019/211(H1N1)pdm09 (Figure 8c).

At the same time, intranasal immunization with a LAIV based on the A/17/Guangdong–Maonan/2019/211(H1N1)pdm09 strain protected 30% more immunized mice from lethal infection with the heterologous A/Indonesia/5/2005(H5N1) IDCDC-RG2 virus on day 6 after immunization. It is worth noting that, unlike the infectious virus adapted to mice, the infectious dose of the A/Indonesia/5/2005(H5N1) IDCDC-RG2 virus was 1 MLD50. A/17/South Africa/2013/01(H1N1)pdm09-based LAIV protected 100% of animals from lethal A/Indonesia/5/2005(H5N1) IDCDC-RG2 infection (Figure 9A). The weight curves also differed insignificantly (Figure 9B). In this case, the viral load was statistically significantly reduced by immunization with A/17/South Africa/2013/01(H1N1)pdm09-based LAIV (Figure 9C).

## 4. Discussion

It has been previously shown in mice that the circulating A/Guangdong–Maonan/SWL1536/2019 (H1N1)pdm09 virus (the recommended reference virus for the 2020–2021 Northern Hemisphere vaccine) is less pathogenic than the A/South Africa/3626/2013 (H1N1)pdm09 virus, which was fatal to mice without any prior adaptation [37]. In addition, there are differences in the polymerase genes between these two viruses. Our study used reassortants containing the same internal gene sequences from a cold-adaptive master donor strain, so the differences in surface glycoproteins may influence the properties of these reassortants, which we attempted to investigate in this study.

A new glycosylation site was identified at position 179 in the HA sequence of the A/Guangdong–Maonan/SWL1536/2019 (H1N1)pdm09 virus. This site was first observed in 2009 and has since disappeared from most strains of pdm09. However, it reappeared in 2016 and became prevalent in seasonal strains by 2020 [38]. In our study of A/17/South Africa/2013/01(H1N1)pdm09, we found that this strain did not have the N179 motif, while (reduction period), while A/17/Guangdong–Maonan/2019/211(H1N1)pdm09 acquired it during the second wave. This is consistent with trends seen globally [39]. According to NetNGlyc predictions and in vivo experiments, introduction of glycan to Asn179 reduces the access of antibodies to these epitopes, as evidenced by decreased neutralizing and hemagglutination titers in vaccinated animals [40,41,42]. In addition, the new glycan alters virus binding to receptors [38].

The process of glycosylation increases the susceptibility of IAVs to soluble and cell-associated lectins. This advantage that IAVs gain from avoiding humoral immunity can be counterbalanced by a weakened infection caused by enhanced innate immune recognition. The presence of a glycan at position 179 leads to the need for compensatory mutations in order to restore the stability of HA trimers. In natural strains, a combination of N179 and I233 is often seen, which prevents the destabilization of β-sheets and allows for the formation of functional trimers [43]. Comparison of the amino acid sequences of A/17/Guangdong–Maonan/2019/211(H1N1)pdm09 with the reference strain A/California/07/2009 (UniProt: C3W5X2_9INFA) confirms the existence of the I233T substitution, which could explain the preservation of replicative capacity even in the face of additional glycan. Thus, the effect of N179 on viral replication and immunogenicity should be considered taking into account the accompanying stabilizing mutations.

The analysis and comparison of putative N-glycosylation sites presented here are solely based on in silico prediction (NetNGlyc) and structural modeling. While an Asn-X-Ser/Thr motif with high jury agreement indicates a position potentially amenable to N-glycosylation, this is not equivalent to demonstrated glycan occupancy. Therefore, functional consequences attributed to the putative glycosides (including Asn179) should be interpreted with caution pending experimental validation.

Our study showed that the vaccine virus A/17/Guangdong–Maonan/2019/211(H1N1)pdm09 was not inferior in productivity to the A/17/South Africa/2013/01(H1N1)pdm09 virus both in CE and in mammalian cell cultures. The vaccine virus A/17/South Africa/2013/01(H1N1)pdm09 replicated better in the lungs of mice. However, we did not find convincing evidence that it actually replicated better in mouse respiratory tracts, as a study of viral load in lungs and nasal passageways after immunization did not show significant differences. Meanwhile, A/17/South Africa/2013/01(H1N1)pdm09 vaccine strain was more prevalent for the formation of virus-specific antibodies, especially at the early stages of immunization compared to A/17/Guangdong–Maonan/2019/211(H1N1)pdm09 vaccine strain.

Changes not only in the HA, but also to the NA may affect the immunogenicity of vaccine strains. Thus, increasing the probability of glycosylation at the NA stem at Asn^42^ site slightly slows down the rate of cleavage of sialic acid and thereby moderately inhibits the release of viruses from upper respiratory tract. The loss of a weak motif at Asn^50^ site partially compensates for the slowdown, allowing replication to continue at a level adequate for antigen presentation [44,45,46]. Since both changes occur outside the catalytic “head” and do not interfere with the main B-cell epitopes of NA, they do not directly reduce immunogenicity [47]. Our previous studies showed that the H6N1/19 virus containing the NA A/Guangdong–Maonan/SWL1536/2019 (H1N1)pdm09 had slightly lower enzymatic activity than the H6N1/13 virus with NA A/South Africa/3626/2013 when tested on a high molecular weight substrate. These changes do not affect cross-reactivity with neuraminidase antibody production in humans who have been vaccinated with seasonal influenza vaccines containing earlier strains [48]. Therefore, modifications to NA in strain/17/Guangdong–Maonan/2019 serve as a mechanism to further attenuate and balance HA-NA, but they are not the primary cause of decreased immunogenicity; instead, the emergence of a new glycosite in HA is responsible for reduced protection against infection [49].

In our experiments, A/17/South Africa/2013/01 induced substantially greater type I IFN and MX1 production in epithelial cells than A/17/Guangdong–Maonan/2019. Because both reassortant vaccine viruses share identical internal genes derived from the same cold-adapted donor (including the NS segment), differences in NS-mediated antagonism of innate signaling are unlikely to account for the observed divergence. It is therefore plausible that alterations in surface glycoproteins (HA and/or NA)—for example, changes in N-glycosylation patterns or epitope accessibility—influence early recognition by innate sensors and downstream signaling. Activation of IFNAR/IFNLR leads to JAK–STAT-dependent induction of interferon-stimulated genes, among which MX1 is a prominent antiviral effector implicated in limiting influenza replication via interactions with viral nucleoprotein [50,51,52,53,54]. A more robust induction of MX1 and other ISGs after exposure to the A/17/South Africa strain may partially explain the enhanced early control of replication and superior short-term protection observed in vivo. Additional mechanistic experiments would be required to define the precise receptor(s) and sensing pathways responsible for these differences; this is beyond the scope of the present study but is an important topic for follow-up work. MX1 is anticipated to impact the severity of viral infection and the extent of harm it causes to different cell types that participate in lymphocyte activation [55].

Our study showed different patterns of type I interferon induction in epithelial and macrophage cells, as well as different production of the MX1 protein by epithelial cells under the influence of the two variants of influenza virus A/H1N1pdm09. THP-1 produced more type I IFN, and A549 produced more MX-1. The difference between the production of MX-1 in epithelial cells and in macrophages when exposed to the influenza virus can be attributed to the different roles they play in the body [56]. Epithelial cells produce MX-1 as part of their innate defense mechanism against viral infections at the point of entry, while macrophages do not activate this mechanism. MX-1 enhances the ability of the body to resist influenza A virus by inducing a rapid inflammatory response in infected respiratory epithelial cells [57]. Epithelial cells may express transcription factor or have MX-1 genes available for rapid induction upon viral detection, while macrophages may not activate these pathways. IFN-α is produced rapidly in response to infections but returns to baseline shortly after. IFN-λ remains for a longer period and is localized to specific tissues. IFNAR is present throughout the body while IFNLR mainly in epithelial cells [50]. This difference in distribution may explain the different responses to infections. Importantly, we did not assess the permissiveness of THP-1 cells for the investigated IAV strains in this study. However, it has been previously shown that these cells do not allow the replication of the influenza virus and, furthermore, the production of the cytokine response does not require the presence of a replicating virus; contact with an inactive virus is sufficient [58].

Another aspect of non-specific immunity is the presence of polyreactive IgM antibodies. These antibodies are a part of the humoral immune system [59]. In previous studies on mice, researchers observed that an effective immune response begins to develop after five days of infection with influenza. This response includes virus-specific IgG antibodies [60]. It has been previously demonstrated that natural polyreactive IgM antibodies produced by B-1 cells exhibit broad cross-reactivity against different influenza viruses. The broad specificity of these antibodies suggests that they may bind to carbohydrates on the surface of the virus [61]. Our study also showed that during the first week after immunization, the A/17/South Africa/2013/01(H1N1)pdm09 virus elicits a strong systemic IgM immune response in the first week following immunization with LAIV. Mean IgM levels decrease over time, while mean IgG levels increase. Although mean local IgA levels before challenge was only slightly higher than those in the unvaccinated control group, local immunity may still contribute to protection shortly after vaccination. Thus, antibody production may also contribute to early protection.

A limitation of the present study is the relatively small number of bases in the cell line studies. At the same time, statistical processing allowed us to identify statistically significant differences.

## 5. Conclusions

Vaccine viruses based on A/South Africa/3626/13 (H1N1)pdm09 or A/Guangdong–Maonan/SWL1536/2019(H1N1)pdm09 showed different glycosylation patterns in their HA. The reproductive activity of the two vaccine strains did not differ in chicken embryos and in mammalian cell culture MDCK. In the A549 cell culture, the LAIV virus A/17/South Africa/2013/01(H1N1)pdm09 caused a more noticeable increase in the production of interferon 1 alpha and MX1 protein compared to the A/17/Guangdong–Maonan/2019/211 (H1N1)pdm09 virus possessing new glycosylation sites including at position 179. Only immunization with LAIV A/17/South Africa/2013/01(H1N1)pdm09 caused a statistically significant increase in IgM and IgG antibodies as soon as in the first week after immunization. Moreover, immunization of mice with the LAIV virus A/17/South Africa/2013/01(H1N1)pdm09 provided 70% protection against lethal infection with 100 MLD50 influenza virus adapted to mice as soon as 6 days after immunization, while immunization with the A/17/Guangdong–Maonan/2019/211 (H1N1)pdm09 vaccine virus did not protect against lethality, but reduced the excretion of the infecting virus from the lungs. But still, immunization with the vaccine virus A/17/Guangdong–Maonan/2019/211(H1N1)pdm09 partially protected mice from the heterologous 1 MLD50 challenge of the influenza A/H5N1 virus, while immunization with the A/17/South Africa/2013/01(H1N1)pdm09 virus provided complete protection.

## Figures and Tables

**Figure 1 vaccines-13-00966-f001:**
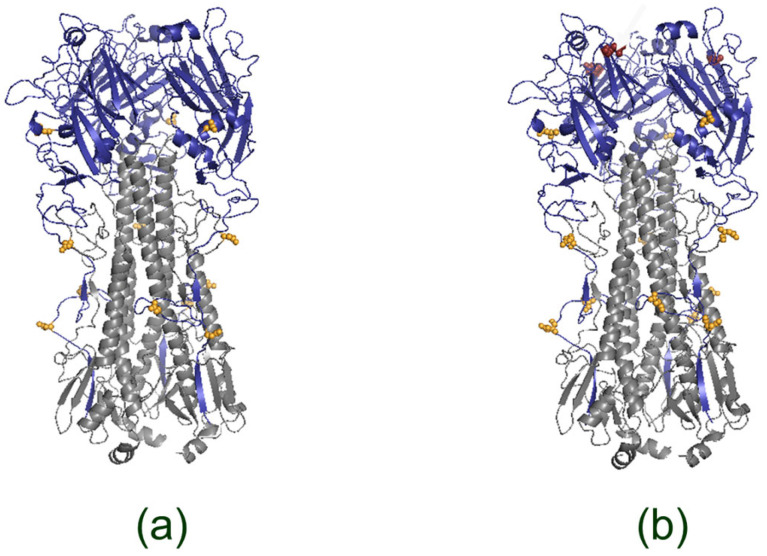
Structural models of HA for LAIV reassortants. HA homotrimers were modeled from amino acid sequences using AlphaFold Server and visualized in PyMOL. (**a**) HA of the cold-adapted reassortant A/17/South Africa/2013/01 (H1N1)pdm09. (**b**) HA of the cold-adapted reassortant A/17/Guangdong–Maonan/2019/211 (H1N1)pdm09. In both panels each monomer is shown with the HA1 globular domain (blue) and HA2 stalk (gray). Predicted N-glycosylation motifs (NetNGlyc) are highlighted in yellow; the putative additional N179 site in the 2019 strain is highlighted in red (panel **b**).

**Figure 2 vaccines-13-00966-f002:**
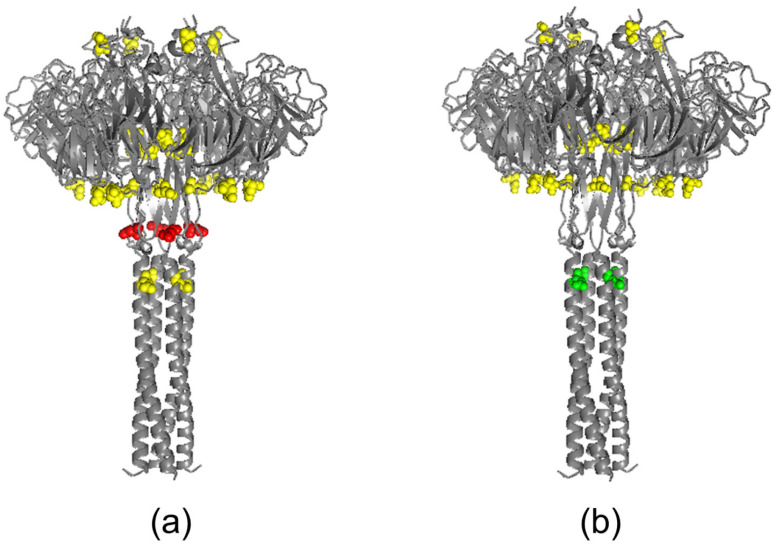
Structural models of NA for LAIV reassortants. NA homotetramers were modeled using AlphaFold Server and visualized in PyMOL. (**a**) A/17/South Africa/2013/01 (H1N1)pdm09. (**b**) A/17/Guangdong–Maonan/2019/211 (H1N1)pdm09. Predicted N-glycosylation motifs (NetNGlyc) are shown in yellow. Specific sites discussed in the text are highlighted N50 in red in panel (**a**); N42 in green in panel (**b**) to indicate predicted loss/gain or altered probability of glycosylation between strains.

**Figure 3 vaccines-13-00966-f003:**
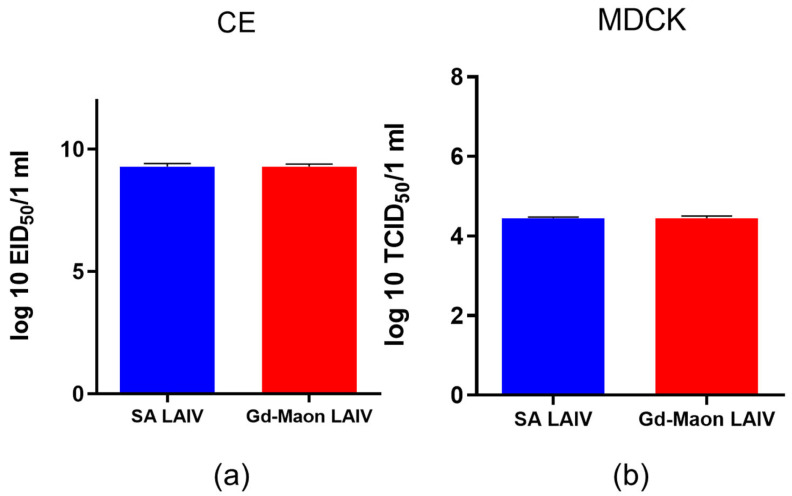
Replication of vaccine viruses in embryonated chicken eggs and MDCK cells. Chicken embryos (CEs) and MDCK cells were infected with ten-fold serial dilutions of each vaccine virus starting from 7 log10 EID50/0.1 mL. (**a**) Virus titers in CEs after incubation at 33 °C for 48 h (EID50). (**b**) Virus titers in MDCK cells after incubation at 33 °C for 72 h (TCID50). Data are presented as mean ± SD (*n* = 5 per group). Statistical differences were assessed by Mann–Whitney test. Error bars represent SD.

**Figure 4 vaccines-13-00966-f004:**
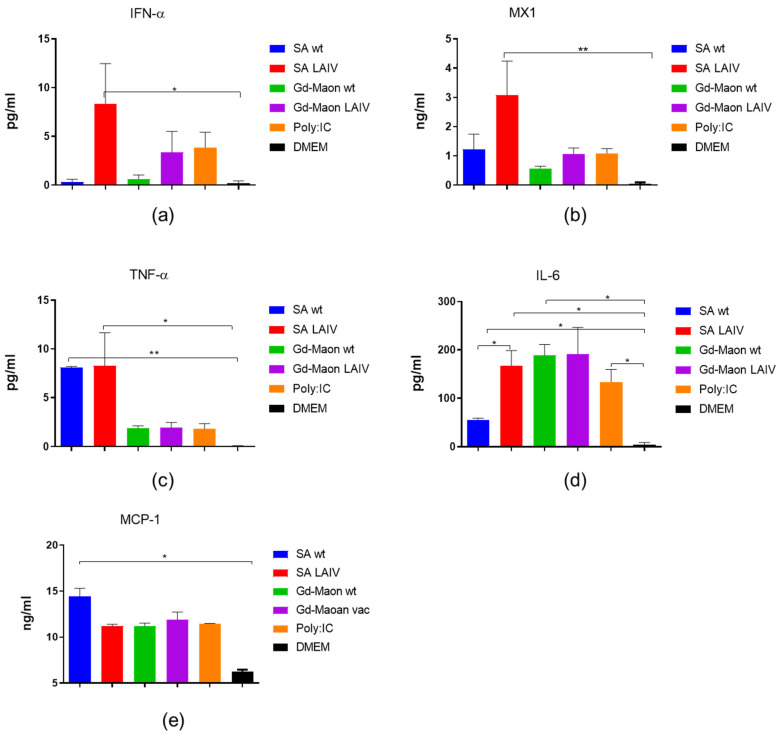
Early cytokine, MCP-1, type I interferon, and MX1 induction in A549 epithelial cells 24 h after infection (ELISA). A549 cells were exposed to indicated viruses and supernatants assayed by ELISA 24 h later. SA wt = A/South Africa/3626/13 (H1N1)pdm09; SA LAIV = A/17/South Africa/2013/01 (H1N1)pdm09; Gd-Maon wt = A/Guangdong–Maonan/SWL1536/2019 (H1N1)pdm09; Gd-Maon LAIV = A/17/Guangdong–Maonan/2019/211 (H1N1)pdm09. Data from two independent experiments performed in duplicate are shown (total *n* = 4). *—*p* < 0.05, **—*p* < 0.01. (**a**) Type I interferon (IFN-α), *p* = 0.0044, Kruskal–Wallis test. (**b**) MX1, *p* = 0.0032, Kruskal–Wallis test. (**c**) TNF-α, *p* = 0.036, Kruskal–Wallis test. (**d**) IL-6, *p* = 0.0035, Kruskal–Wallis test. (**e**) MCP-1, *p* = 0.0034, Kruskal–Wallis test.

**Figure 5 vaccines-13-00966-f005:**
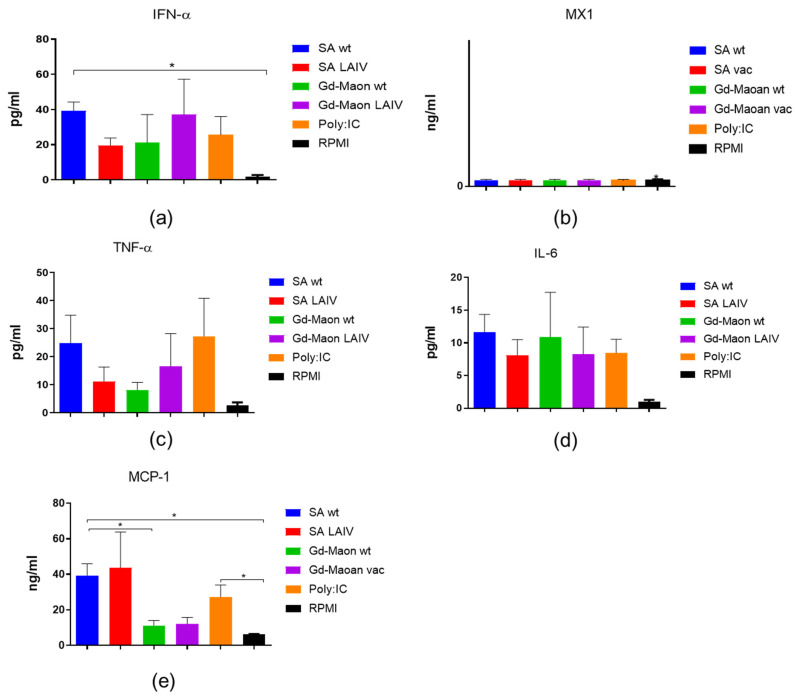
Early cytokine, interferon and chemokine responses in THP-1 macrophage-like cells 24 h after infection (ELISA). THP-1 cells were infected and supernatants assayed by ELISA. Strain notation as in Figure 4. Data represent three independent experiments (total *n* = 6 per condition). *—*p* < 0.05. (**a**). *p* = 0.047, Kruskal–Wallis test. (**b**). *p* > 0.999, Kruskal–Wallis test. (**c**). *p* = 0.13, Kruskal–Wallis test. (**d**). *p* = 0.26, Kruskal–Wallis test. (**e**). *p* = 0.031, Kruskal–Wallis test. (**a**) Type I interferon (IFN-α), *p* = 0.047, Kruskal–Wallis test. (**b**) MX1, *p* > 0.999, Kruskal–Wallis test. (**c**) TNF-α, *p* = 0.13, Kruskal–Wallis test. (**d**) IL-6, *p* = 0.26, Kruskal–Wallis test. (**e**) MCP-1, *p* = 0.031, Kruskal–Wallis test.

**Figure 6 vaccines-13-00966-f006:**
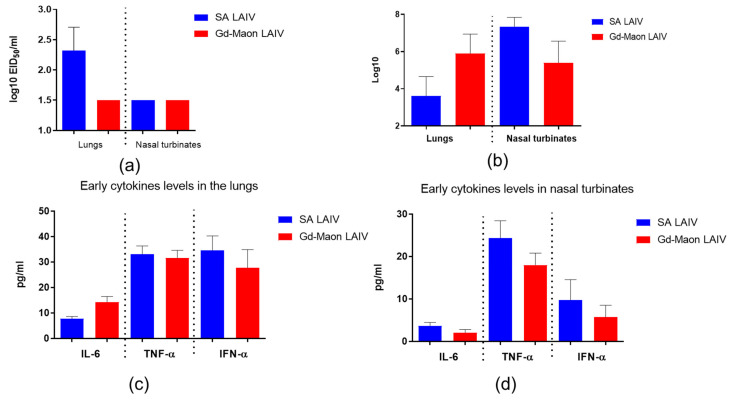
Vaccine virus replication and early cytokine production in mouse respiratory tract on day 3 post-immunization. Mice were immunized intranasally and tissues collected on day 3. (**a**) Infectious virus titers in lung homogenates determined by egg titration (EID50); *n* = 6. Kruskal–Wallis test, *p* = 0.0019. (**b**) Viral RNA load in lungs and nasal turbinates measured by rRT-PCR; *n* = 6. Kruskal–Wallis test, *p* = 0.013. (**c**) Early cytokine levels in lung homogenates measured by ELISA; *n* = 6 (Kruskal–Wallis, *p* > 0.05). (**d**) Early cytokine levels in nasal turbinates measured by ELISA; *n* = 6 (Kruskal–Wallis, *p* > 0.05). Data are mean ± SD.

**Figure 7 vaccines-13-00966-f007:**
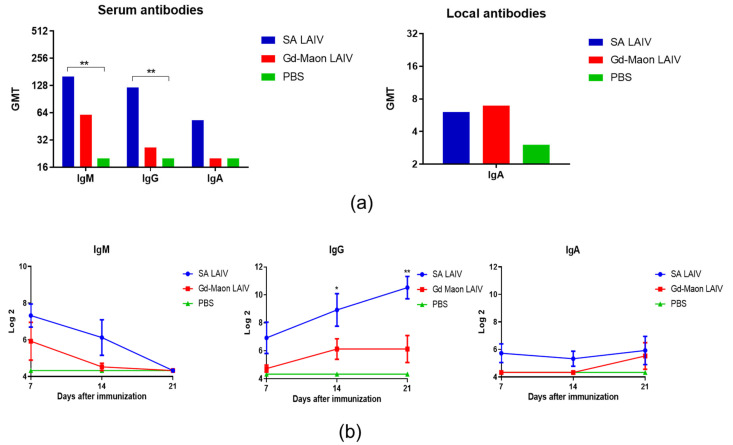
Antibody response to A/California/09/07(H1N1)pdm09 after immunization of mice (*n* = 5). Sera and mucosal samples were collected from *n* = 6 mice per group. *—*p* < 0.05; **—*p* < 0.01. (**a**) Geometric mean titers (GMTs) of serum (IgM, IgG) and local (IgA) antibodies on day 6 post-immunization. (**b**) Dynamics of serum antibody titers over 3 weeks after immunization.

**Figure 8 vaccines-13-00966-f008:**
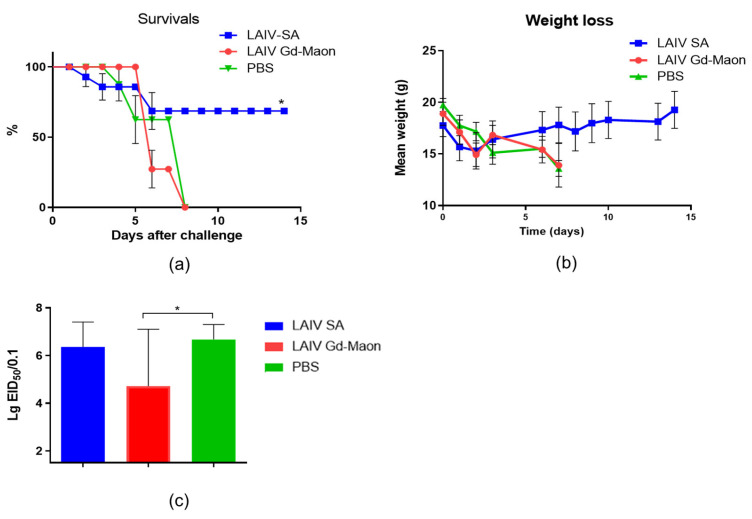
Protection against homologous challenge with A/California/09/07 (H1N1)pdm09-MA (100 MLD50). Mice (*n* = 10 per group) were challenged on day 6 post-immunization. Data from one of two independent experiments are shown. (**a**) Survival curves (Log-rank Mantel–Cox test; * *p* < 0.05 vs. PBS). (**b**) Body-weight changes post-challenge (*n* = 10; Mann–Whitney test). (**c**) Infectious virus titers in the lungs on day 3 after challenge (*n* = 5; Mann–Whitney test; * *p* < 0.05 vs. PBS).

**Figure 9 vaccines-13-00966-f009:**
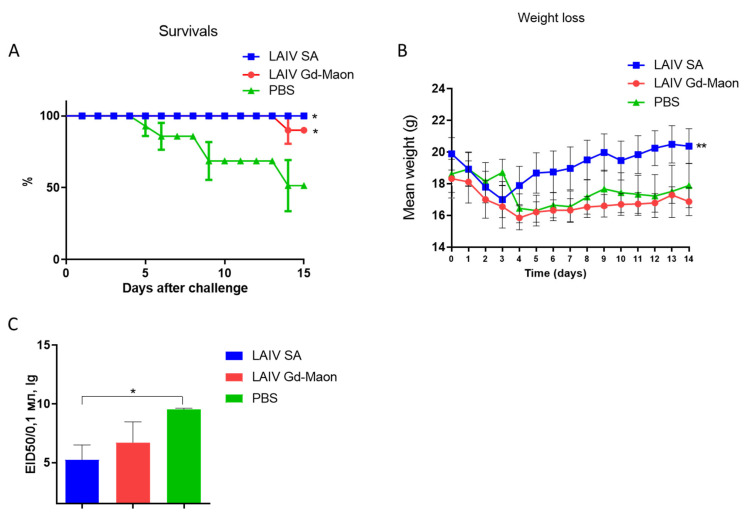
Cross-protection against heterologous A/Indonesia/5/2005 (H5N1) IDCDC-RG2 challenge (1 MLD50). Mice (*n* = 10 per group) were challenged and monitored. Data from one of two independent experiments are shown. (**A**) Survival curves (Log-rank Mantel–Cox test; * *p* < 0.05 vs. PBS). (**B**) Body-weight changes post-challenge (*n* = 10; Kruskal–Wallis test; ** *p* < 0.01 vs. PBS). (**C**) Infectious virus titers in the lungs on day 3 after challenge (*n* = 5; Mann–Whitney test; * *p* < 0.05 vs. PBS). SA LAIV provided complete protection; Maonan LAIV provided partial protection.

**Table 1 vaccines-13-00966-t001:** Predicted N-glycosylation motifs in the HA protein of A(H1N1)pdm09 viruses from 2013 and 2019.

Position	Motif	2013 Strain: Glycosylation	2019 Strain: Glycosylation	Observed Changes
27	NNST	–	–	No changes
28	NSTD	+++	+++	No changes
40	NVTV	++	++	No changes
104	NGTC	+	+	No changes
179	NQTY	Absent	++	New site in 2019
293	NTTC	–	–	No changes
304	NTSL	++	++	No changes
498	NGTY	+	+	No changes
557	NGSL	++	++	No changes

Symbols reflect NetNGlyc 1.0 confidence: “Absent” = no sequon; “–” = potential < 0.50 (not predicted); “+” = 0.50–<0.75; “++” = 0.75–<0.90; “+++” ≥0.90.

**Table 2 vaccines-13-00966-t002:** Predicted N-glycosylation motifs in the NA protein of A(H1N1)pdm09 viruses from 2013 and 2019.

Position	Motif	2013 Strain: Glycosylation	2019 Strain: Glycosylation	Observed Changes
42	NQSQ	+	++	Probability enhancement
50	NQSV	+	–	Vanishing (2019)
58	NNTW	+	+	No changes
63	NQTY	++	++	No changes
68	NISN	++	+	Minor decrease
88	NSSL	++	++	No changes
146	NGTI	++	++	No changes
235	NGSC	++	++	No changes
386	NFSI	–	–	No changes

Symbols reflect NetNGlyc 1.0 confidence: “Absent” = no sequon; “–” = potential < 0.50 (not predicted); “+” = 0.50–<0.75; “++” = 0.75–<0.90.

## Data Availability

All necessary data are contained in the text of the manuscript, tables and figures.

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
