# Peer review of "Differences in Glycoproteins and the Potential for Early Protection Using LAIV Based on Drift Variants of the A/H1N1pdm09 Influenza Virus"

_vaccines, 2025, doi:10.3390/vaccines13090966_

Round 1
Reviewer 1 Report
Comments and Suggestions for Authors
Authors studied two influenza strains for live attenuated vaccine. Despite the inactived vaccines are more popular, the knowlidge of the changes in the surface proteins and immune induced activity is actual and useful for global health.
Line 113: Please. Indicate the names of two drift LAIV vaccine strains of A/H1N1pdm09.
Line 114-127. It is not clearly, why you mention this information here. Please, show the actuality and necessary of this information for your study. It might be more suitable to use this fragment in methods section.
Objective of your work should be in the end of the introduction. There is no information in the objective about structure of glycoproteins as presented in the title of the manuscript.
Line 132: It would be more delicately say “candidate to vaccine strains”.
Table 1 and 2. Please, indicate the meaning of -, +, ++, +++ and others valuables in the tables.
Figure 4 shows the production of TNF and IL-6, except type 1 interferon and MX1. Please, correct this information.
Why didn't you study reproduction of MCP-1 in A549 cells?
Figure 6. Please, clarify the meaning of "(n=5 or 6)" and give name and description to picture B. Here the description of picture B and C corresponds to C and D.
Figure 8. Please, change A and B description. The same in the figure 9.
Information in lines 430-440, 442-458 could be better located in the introduction.
Line 379 - 383: please, give a reference.
The Discussion section has much information that could be located in the Introduction. Also, there are many self citations. Could you please expand your look at studies of other authors similar to yours.
Suggest to add to the list of abbreviations the following short names: IL-6, MCP-1, TNF.
Author Response
Comments and Suggestions for Authors
Authors studied two influenza strains for live attenuated vaccine. Despite the inactived vaccines are more popular, the knowlidge of the changes in the surface proteins and immune induced activity is actual and useful for global health.
Line 113: Please. Indicate the names of two drift LAIV vaccine strains of A/H1N1pdm09.
Response. Thank you for this comment. We have added virus names starting from the abstract.
Line 114-127. It is not clearly, why you mention this information here. Please, show the actuality and necessary of this information for your study. It might be more suitable to use this fragment in methods section.
Response: We thank the Reviewer for this valuable comment. In the revised version, we shortened and reformulated this fragment to make its relevance clearer and to better link it with the rationale of our study. The text now emphasizes why this background information is necessary for the reader and how it supports the objectives of our work. We believe this modification improves the logical flow of the Introduction and addresses the Reviewer’s concern.
Objective of your work should be in the end of the introduction. There is no information in the objective about structure of glycoproteins as presented in the title of the manuscript.
Response: We fully agree with the Reviewer. The Objective statement has now been relocated to the end of the Introduction. Moreover, we have revised the wording of the objective to explicitly include the structural analysis of viral glycoproteins, thereby ensuring consistency with the manuscript title.
Line 132: It would be more delicately say “candidate to vaccine strains”
Response. The authors thank the Reviewer for this helpful suggestion. We agree that candidate vaccine strains is more precise and have amended the manuscript accordingly.
Table 1 and 2. Please, indicate the meaning of -, +, ++, +++ and others valuables in the tables.
Response. The authors thank the Reviewer for this suggestion. We have added a legend to Tables 1 and 2 explaining the meaning of “Absent”, “–”, “+”, “++”, and “+++”
Figure 4 shows the production of TNF and IL-6, except type 1 interferon and MX1. Please, correct this information.
Response: We thank the Reviewer for this careful reading and for raising the point. We have corrected the figure legend.
Why didn't you study reproduction of MCP-1 in A549 cells?
Response: We thank the Reviewer for this pertinent question. We have added the missing data.
Figure 6. Please, clarify the meaning of "(n=5 or 6)" and give name and description to picture B. Here the description of picture B and C corresponds to C and D.
Response: We thank the reviewer for this valuable comment. We have clarified the figure legend to avoid confusion. In panel A, viral titers were determined in chicken embryos (n=6). In panels B–D, experiments were performed in immunized mice (n=6). The figure legend has been corrected accordingly, and panel labels (B–D) have been revised.
Figure 8. Please, change A and B description. The same in the figure 9.
Response: We thank the reviewer for pointing this out. The descriptions of panels A and B in Figures 8 and 9 have been corrected accordingly.
Information in lines 430-440, 442-458 could be better located in the introduction.
Response: We appreciate the Reviewer’s suggestion. Accordingly, the indicated fragment has been moved from the Discussion section to the Introduction, where it provides a stronger background and contextual framework for the study. We believe this change has improved the overall clarity and structure of the manuscript.
Line 379 - 383: please, give a reference.
Response: We are grateful for this comment. We have now provided supporting references for the statement at lines 379-383.
The Discussion section has much information that could be located in the Introduction. Also, there are many self citations. Could you please expand your look at studies of other authors similar to yours.
Response: We are grateful for this comment. We have rewritten the discussion and added links from other authors.
Suggest to add to the list of abbreviations the following short names: IL-6, MCP-1, TNF.
Response: We thank the reviewer for the suggestion. The abbreviations IL-6, MCP-1, and TNF have been added to the list of abbreviations.
Reviewer 2 Report
Comments and Suggestions for Authors
The manuscript addresses an interesting and relevant question in influenza vaccinology: how glycoprotein structural variations in drifted A(H1N1)pdm09 strains impact early protective immunity and cross-protection when used in live attenuated influenza vaccines (LAIV). The focus on early immune responses, glycosylation patterns, and heterologous protection is conceptually valuable and has potential novelty. However, the work is weakened by issues in experimental design, clarity of hypothesis, data interpretation, and language quality.
- The title does not cover the research performed since it does not include the evaluation of trained immunity as it is written. The authors must improve it.
- Lines 63-65 must be rewritten.
- Lines 74-76 must be rewritten. This is not an adequate description of trained immunity since myeloid cells are part of the innate immune system and are relevant for this phenomenon.
- Line 90: CD8+.
- Lines 11-127 must be rewritten since they are not well redacted.
- Line 180: 1x106.
- Myxovirus resistance 1 (MX1) relevance must be indicated in the introduction. Also, avoid the wrong use of uppercase.
- The authors must justify the use of the mouse strain and the sex.
- There is no rationale exposed to evaluate the brain. The authors are inconsistent with the background presented and the methodology.
- MLD50 is not defined, and the dose needs to be justified.
- The authors must experimentally probe the differences in glycosylations.
- Line 273: TNF-α.
- All figure titles must indicate the conclusion of the figure.
- All figures must indicate the experimental N, the independent experiments performed, and the statistical analysis.
- Lines 296-303 must be rewritten for clarity.
- It is not clear if the IAV can infect the THP-1 cells or not. This point is relevant to understanding how IAV can promote the cytokine secretion. It is not well connected to the rest of the experiments.
- The legend of figure 6 must be located down the figure. Also, the authors mentioned P= 0.0019, Kruskal-Wallis test, but no statistical difference is shown in the figure.
- Figure 7: Antibody titers must be shown in concentration.
- Trained immunity is not evaluated properly.
- The experimental design is not adequate for this study.
The quality and grammar must be improved.
Author Response
The manuscript addresses an interesting and relevant question in influenza vaccinology: how glycoprotein structural variations in drifted A(H1N1)pdm09 strains impact early protective immunity and cross-protection when used in live attenuated influenza vaccines (LAIV). The focus on early immune responses, glycosylation patterns, and heterologous protection is conceptually valuable and has potential novelty. However, the work is weakened by issues in experimental design, clarity of hypothesis, data interpretation, and language quality.
The title does not cover the research performed since it does not include the evaluation of trained immunity as it is written. The authors must improve it.
Response: We are grateful for this comment. We have changed the name.
Lines 74-76 must be rewritten. This is not an adequate description of trained immunity since myeloid cells are part of the innate immune system and are relevant for this phenomenon.
Response: We thank the reviewer for this important remark. We agree that the original phrasing was imprecise and could be misinterpreted regarding the cell types involved in trained immunity. We have revised lines 74–76 to clarify that trained immunity refers to a functional reprogramming of innate immune cells and their progenitors. We hope this correction addresses the reviewer’s concern.
Line 90: CD8+.
Response: We thank the reviewer for pointing out these typographical errors. The manuscript has been updated accordingly.
Lines 11-127 must be rewritten since they are not well redacted.
Response: We thank the Reviewer for the valuable comment. The section covering lines 11–127 has been carefully revised to improve clarity, style, and overall readability. We believe the revised version is now more concise and consistent with academic writing standards.
Line 180: 1x106.0
Response: We thank the reviewer for pointing out these typographical errors. The manuscript has been updated accordingly.
Myxovirus resistance 1 (MX1) relevance must be indicated in the introduction. Also, avoid the wrong use of uppercase.
Response: We thank the reviewer for this helpful suggestion. We have added a brief explanation of the relevance of MX1 to the Introduction.
The authors must justify the use of the mouse strain and the sex.
Response: We thank the reviewer for this helpful comment. We have added a short justification to the Methods (section 2.4, “Mice”) explaining our choice of strain, sex and age.
There is no rationale exposed to evaluate the brain. The authors are inconsistent with the background presented and the methodology.
Response: We thank the Reviewer for flagging this inconsistency. To clarify: brain/CNS tissues were not collected in this study — the previous reference to brain sampling was an editorial oversight and has been removed; all mentions of brain collection have been deleted from the Methods.
MLD50 is not defined, and the dose needs to be justified.
Response: Thank you for this important comment. We have described how MLD50 values were obtained: preliminary titration experiments in naive CBA mice (groups of five animals per ten-fold serial dilution, 50 µL per mouse, observed for 14 days) and calculation by the Reed–Muench method.
The authors must experimentally probe the differences in glycosylations.
Response: We thank the Reviewer for this important suggestion. We agree that in silico prediction alone does not prove site occupancy or glycan structure. In the revised manuscript we have explicitly stated that all glycosylation assignments are predictive and unconfirmed experimentally.
Line 273: TNF-α.
Response: We thank the reviewer for pointing out these typographical errors. The manuscript has been updated accordingly.
All figure titles must indicate the conclusion of the figure. All figures must indicate the experimental N, the independent experiments performed, and the statistical analysis.
Response: Thank you for excellent point. We have made the requested changes throughout the manuscript.
Lines 296-303 must be rewritten for clarity. It is not clear if the IAV can infect the THP-1 cells or not. This point is relevant to understanding how IAV can promote the cytokine secretion. It is not well connected to the rest of the experiments.
Response: We thank the reviewer for raising this important point. We referred in the discussion to existing work on this issue.
The legend of figure 6 must be located down the figure. Also, the authors mentioned P= 0.0019, Kruskal-Wallis test, but no statistical difference is shown in the figure.
Response: We thank the reviewer for this comment. The reported P value of .00019 shows differences between all groups studied by multiple comparisons. Pairwise comparisons showed no differences between individual groups. We explained this in the text.
Figure 7: Antibody titers must be shown in concentration.
Response: We thank the reviewer for this comment. The tests at our disposal allow us to determine antibody titers expressed as the final serum dilution giving an optical density exceeding the control well by 3 standard deviations, which is generally accepted. This is indicated in the materials and methods.
Trained immunity is not evaluated properly.
Response: We thank the reviewer for this criticism. Indeed, it is difficult to cover all aspects of innate immunity in a single study after immunization with an intranasal vaccine. However, we hope that we will be able to draw attention to the differences that we have shown, which may contribute to a better understanding of the mechanisms of early protection.
The experimental design is not adequate for this study.
Response: We thank the reviewer for this comment. We have revised the description of the study design for greater clarity.
Reviewer 3 Report
Comments and Suggestions for Authors
The authors evaluated the immune response and protection efficacy of two live attenuated influenza vaccines (LIAV) with HA and NA gene donors from strains A/South Africa/3626/2013 (H1N1) pdm09 and A/Guangdong-Maonan/SWL1536/2019 (H1N1) pdm09. They found that the glycosylation patterns of HA and NA in the LIAV strains influenced early innate and adaptive immune responses, as well as cross-protective efficacy. However, further evidence is needed to supported this conclusion.
1.Since HA genes from strains A/South Africa/3626/13 (H1N1) pdm09 and A/Guangdong-Maonan/SWL1536/2019 (H1N1) pdm09 exhibit differences in glycosylation patterns and antigenic epitopes, it is recommended to use the HA and NA gene from strain A/South Africa/3626/13 (H1N1) pdm09 and modify the glycosylation patterns to match those of strain A/Guangdong-Maonan/SWL1536/2019 (H1N1) pdm09, followed by a comparative analysis of their immune response and protection efficacy.
2.The antibody response induced by LAIVs was determined against A/ California/09/07(H1N1)pdm09. However, due to potential antigenic differences between the detection antigen and strains A/South Africa/3626/13 (H1N1) pdm09 and A/Guangdong-Maonan/SWL1536/2019 (H1N1) pdm09, the antibody response should be detected by different antigens.
3.There were numerous inconsistencies and errors in the manuscript, for example, the strain names mentioned in the abstract differed from those used in the main text; line 284 the figure was not properly cited in text; line312, the figure legend did not correspond accurately with the figure itself; etc.
4.The methods section didn’t include the procedure for sample collection and antibody detection.
5. Vaccination of the A/17/Guangdong–Maonan/2019/211(H1N1)pdm09 strain did not confer protection against lethal infection with the mouse-adapted A/H1N1pdm09 virus. In contrast, vaccination of the A/17/South Africa/2013/01(H1N1)pdm09 protected 70% of the animals from mortality, However, a significant reduction in viral load was observed only in the A/17/Guangdong–Maonan/2019/211(H1N1)pdm09 group. The results warrant further interpretation.
Author Response
Comments and Suggestions for Authors
The authors evaluated the immune response and protection efficacy of two live attenuated influenza vaccines (LIAV) with HA and NA gene donors from strains A/South Africa/3626/2013 (H1N1) pdm09 and A/Guangdong-Maonan/SWL1536/2019 (H1N1) pdm09. They found that the glycosylation patterns of HA and NA in the LIAV strains influenced early innate and adaptive immune responses, as well as cross-protective efficacy. However, further evidence is needed to supported this conclusion.
1.Since HA genes from strains A/South Africa/3626/13 (H1N1) pdm09 and A/Guangdong-Maonan/SWL1536/2019 (H1N1) pdm09 exhibit differences in glycosylation patterns and antigenic epitopes, it is recommended to use the HA and NA gene from strain A/South Africa/3626/13 (H1N1) pdm09 and modify the glycosylation patterns to match those of strain A/Guangdong-Maonan/SWL1536/2019 (H1N1) pdm09, followed by a comparative analysis of their immune response and protection efficacy.
Response: We thank the Reviewer for this excellent suggestion. We agree that the most rigorous way to test the causal effect of individual N-glycosylation sites is to use isogenic constructs that differ only in the presence/absence of the glycan sequon (site-directed mutants). Generation and in vivo testing of such mutants is planned as a follow-up study.
2.The antibody response induced by LAIVs was determined against A/ California/09/07(H1N1)pdm09. However, due to potential antigenic differences between the detection antigen and strains A/South Africa/3626/13 (H1N1) pdm09 and A/Guangdong-Maonan/SWL1536/2019 (H1N1) pdm09, the antibody response should be detected by different antigens.
Response: We thank the Reviewer for this important point. In the present study we used A/California/09/07(H1N1)pdm09 as a reference antigen to assess cross-reactive antibody responses across groups. We agree that measurement of antibody titers specifically against the HA/NA of A/South Africa/3626/13 and A/Guangdong-Maonan/SWL1536/2019 would provide a more precise view of homologous and strain-specific responses. These additional assays are planned and will be reported in a follow-up manuscript.
3.There were numerous inconsistencies and errors in the manuscript, for example, the strain names mentioned in the abstract differed from those used in the main text; line 284 the figure was not properly cited in text; line312, the figure legend did not correspond accurately with the figure itself; etc.
Response: Thank you for this careful reading and for pointing out the inconsistencies. We have reviewed the whole manuscript and corrected the errors you indicated.
4.The methods section didn’t include the procedure for sample collection and antibody detection.
Response: Thank you for this comment. We have added a new subsection (Sample collection and antibody detection) to the Materials and Methods that provides full details of sample collection (time points, sampling procedures, sample processing and storage) and the ELISA procedure used to measure serum and mucosal antibodies.
- Vaccination of the A/17/Guangdong–Maonan/2019/211(H1N1)pdm09 strain did not confer protection against lethal infection with the mouse-adapted A/H1N1pdm09 virus. In contrast, vaccination of the A/17/South Africa/2013/01(H1N1)pdm09 protected 70% of the animals from mortality, However, a significant reduction in viral load was observed only in the A/17/Guangdong–Maonan/2019/211(H1N1)pdm09 group. The results warrant further interpretation.
Response: Thank you for this comment. These results may indicate that even when a vaccine fails to protect against lethality, this does not mean that there is no protection at all. This protection is simply expressed in a decrease in the infectious titer of the virus in the lungs. Considering that the high multiplicity of infection that we used in the mouse experiment does not occur in natural infection, the data on protection obtained in mice are encouraging with respect to human protection.
Reviewer 4 Report
Comments and Suggestions for Authors
This work makes a good impression. Positively assessing the work as a whole, I would like to express a few suggestions and questions to the authors.
- The Asn-X-Ser/Thr motif is a typical motif for detecting glycosylation sites, which is certainly not a general rule. A link to the source of this pattern would improve the understanding of the work presented.
- Was the detection of the glycosylation site performed only by theoretical methods or confirmed experimentally (for example, by radioimmunoprecipitation analysis)?
Author Response
Comments and Suggestions for Authors
This work makes a good impression. Positively assessing the work as a whole, I would like to express a few suggestions and questions to the authors.
- The Asn-X-Ser/Thr motif is a typical motif for detecting glycosylation sites, which is certainly not a general rule. A link to the source of this pattern would improve the understanding of the work presented.
- Was the detection of the glycosylation site performed only by theoretical methods or confirmed experimentally (for example, by radioimmunoprecipitation analysis)?
We thank Reviewer for the helpful comments.
- We agree that the canonical N-linked glycosylation consensus motif should be cited and that its limitations be made explicit in the manuscript. We have added a sentence in the Introduction clarifying that the canonical motif for co-translational N-glycosylation is Asn-X-Ser/Thr and we cite a mechanistic review and a contemporary survey of atypical glycosylation.
- Regarding experimental confirmation, we confirm that the glycosylation analysis presented in this manuscript was entirely in silico and that no biochemical or mass-spectrometric validation was performed. We have amended the Methods and Discussion to state this limitation explicitly.
Round 2
Reviewer 2 Report
Comments and Suggestions for Authors
The revised version demonstrates an effort to address the comments raised during the first round of review. The manuscript now provides a more organized narrative, with clearer transitions between the background, aims, methods, and results. The inclusion of additional details on experimental design and statistical approaches improves transparency, and the English language has been edited to reduce redundancy and improve readability. However, while progress is evident, several central issues remain unresolved or only superficially corrected, limiting the overall strength of the work.
Despite improvements, the core scientific limitations persist, like the lack of functional validation for glycosylation effects, small sample sizes, incomplete control groups, and occasional overinterpretation of correlative data. The figures, though clearer, still do not consistently communicate results effectively. Finally, the manuscript remains overly long, with repeated emphasis on well-known aspects of influenza vaccinology that detract from its novelty.
Also, THP-1 assays may provide a preliminary screen, but their predictive value for vaccine-induced innate immunity is weak. Using them as a central pillar of mechanistic explanation, as the manuscript currently does, is not logically solid.
Additionally, in in vivo experiments the authors do not show all the evidence needed to probe the infection, like immune cell infiltration in BAL or lungs. Importantly, there is not an analysis of the innate immune response to allow linking with the observed in vitro studies.
Previously, it was required that the authors indicate the independent experiments performed, since it is not the same to do it once with 6 mice as two times with 3 mice each.
All these points need to addressed.
Comments on the Quality of English LanguageMinor revision required.
Author Response
The revised version demonstrates an effort to address the comments raised during the first round of review. The manuscript now provides a more organized narrative, with clearer transitions between the background, aims, methods, and results. The inclusion of additional details on experimental design and statistical approaches improves transparency, and the English language has been edited to reduce redundancy and improve readability. However, while progress is evident, several central issues remain unresolved or only superficially corrected, limiting the overall strength of the work.
Despite improvements, the core scientific limitations persist, like the lack of functional validation for glycosylation effects, small sample sizes, incomplete control groups, and occasional overinterpretation of correlative data. The figures, though clearer, still do not consistently communicate results effectively. Finally, the manuscript remains overly long, with repeated emphasis on well-known aspects of influenza vaccinology that detract from its novelty.
Response. We thank the respected reviewer for these very valuable comments. We have added a note at the end of the text about the limitations of the study. We also agree that the sections are possibly extended and of a didactic nature. On the other hand, we have tried to summarize the studies conducted on this topic as much as possible, which can help further research. We are convinced that studies of innate immunity and the connection with the properties of vaccine antigens will be continued also in connection with the expanding spectrum of pathogens against which vaccines need to be created.
Also, THP-1 assays may provide a preliminary screen, but their predictive value for vaccine-induced innate immunity is weak. Using them as a central pillar of mechanistic explanation, as the manuscript currently does, is not logically solid.
Response. We thank the reviewer for this criticism and agree that screening on macrophages alone cannot provide a complete characterization of the immune response. However, it is worth considering that macrophages are non-permissive for influenza viruses, and inactivated viruses or peptides can be used to assess the cytokine profile of drugs. In this regard, the data obtained on this cell line still seem significant to us.
Additionally, in in vivo experiments the authors do not show all the evidence needed to probe the infection, like immune cell infiltration in BAL or lungs. Importantly, there is not an analysis of the innate immune response to allow linking with the observed in vitro studies.
Response. We thank the reviewer for this criticism. Indeed, LAIV stimulates T cell responses by replicating in the nasopharyngeal epithelial cells, where viral proteins are presented on major histocompatibility complexes, activating virus-specific T cells which contribute to heterosubtypic immunity and provide protection against a broader spectrum of influenza viruses. We have very preliminary data that there is no convincing evidence of cellular immunity in the phenomenon of early protection within 5 days after immunization with live mucosal vaccine. However, this data must be verified.
Previously, it was required that the authors indicate the independent experiments performed, since it is not the same to do it once with 6 mice as two times with 3 mice each.
Response. We thank the reviewer for this criticism. We have added in the figure legends when data from independent experiments are presented. In the experiment on the isolation of vaccine virus from the lungs and the induction of cytokines in mice, the number of repetitions is the number of mice, and there are more than three.
In addition, we had professional editing of the English language.
Reviewer 3 Report
Comments and Suggestions for Authors
Vaccination of the A/17/Guangdong–Maonan/2019/211(H1N1)pdm09 strain did not confer protection against lethal infection with the mouse-adapted A/H1N1pdm09 virus, while vaccination of the A/17/South Africa/2013/01(H1N1)pdm09 protected 70% of the animals from mortality. However, a significant reduction in viral load was observed in the A/17/Guangdong–Maonan/2019/211(H1N1)pdm09 group,not in the A/17/South Africa/2013/01(H1N1)pdm09 group, why?
Author Response
Vaccination of the A/17/Guangdong–Maonan/2019/211(H1N1)pdm09 strain did not confer protection against lethal infection with the mouse-adapted A/H1N1pdm09 virus, while vaccination of the A/17/South Africa/2013/01(H1N1)pdm09 protected 70% of the animals from mortality. However, a significant reduction in viral load was observed in the A/17/Guangdong–Maonan/2019/211(H1N1)pdm09 group,not in the A/17/South Africa/2013/01(H1N1)pdm09 group, why?
Response. We thank the reviewer for this remark. This may be due to the fact that in this particular case LAIV reduces the inflammatory response at this time point, but in this case it is not enough to prevent the death.
Round 3
Reviewer 2 Report
Comments and Suggestions for Authors
The revised manuscript shows minor improvements in clarity and figure presentation, but the core scientific limitations remain unresolved. The central weakness is the lack of functional validation for glycosylation effects, leaving the main hypothesis speculative. Sample sizes are still small, with no evidence of power calculations or independent replicates, and the authors did not clarify whether animal data come from single or multiple experiments, raising concerns about reproducibility. Control groups are incomplete, making it difficult to attribute observed effects specifically to glycoprotein changes. Correlative data from A549 and THP-1 cells continue to be overinterpreted, with the latter treated as mechanistic evidence despite its poor predictive value. Figures, while slightly clearer, still lack consistency and direct statistical annotation, and the manuscript remains overly long and repetitive, diluting its novelty. Finally, the in vivo model is insufficiently characterized, as key evidence such as immune cell infiltration or innate immune analysis in BAL and lungs is missing, leaving no clear link between the in vitro observations and the animal outcomes.
Overall, despite editing and presentation improvements, the revised version still does not adequately resolve the key concerns. The manuscript remains weakened by speculative mechanistic claims, insufficient validation, limited controls, and incomplete immunological analysis.
Comments on the Quality of English LanguageThy quality has been improved.
Author Response
On the one hand, the respected reviewer is right in noting the shortcomings of the study. On the other hand, we directly indicated these limitations in the text of the article. The authors do not insist on the exhaustive nature of their conclusions. But we have facts - two existing strains of one subtype, which have completely different patterns of protection in the early stages after immunization. These observations are repeated from experiment to experiment, and we cannot simply brush them aside. If these strains also differed in reproduction, then one could conclude that one strain reproduces better and protects better, or vice versa. But in our case, there are no differences in reproductive activity, no matter how we studied them - on eggs, on cells and on mice. Yes, at the moment we can only speculate on what differences are involved in the effect of different protection. But I will repeat once again that the authors do not claim the comprehensive nature of the observations obtained.